# The Association between Australian Childcare Centre Healthy Eating Practices and Children’s Healthy Eating Behaviours: A Cross-Sectional Study within Lunchbox Centres

**DOI:** 10.3390/nu13041139

**Published:** 2021-03-30

**Authors:** Courtney Barnes, Sze Lin Yoong, Luke Wolfenden, Nicole Nathan, Taya Wedesweiler, Jayde Kerr, Nicole Pearson, Alice Grady

**Affiliations:** 1Hunter New England Population Health, Wallsend, NSW 2287, Australia; Serene.Yoong@health.nsw.gov.au (S.L.Y.); luke.wolfenden@health.nsw.gov.au (L.W.); Nicole.nathan@health.nsw.gov.au (N.N.); taya.wedesweiler@health.nsw.gov.au (T.W.); Jayde.Kerr@health.nsw.gov.au (J.K.); nicole.pearson@health.nsw.gov.au (N.P.); alice.grady@health.nsw.gov.au (A.G.); 2School of Medicine and Public Health, The University of Newcastle, Callaghan, NSW 2308, Australia; 3Hunter Medical Research Institute, New Lambton, NSW 2305, Australia; 4Priority Research Centre for Health Behaviour, The University of Newcastle, Callaghan, NSW 2308, Australia; 5School of Health Sciences, Swinburne University of Technology, Melbourne, VIC 3122, Australia

**Keywords:** nutrition, healthy eating, childcare, policy, practice, lunchbox, preschool

## Abstract

The association between healthy eating practices and child dietary intake in childcare centres where parents pack foods from home has received little attention. This study aimed to: (1) Describe the nutritional content of foods and beverages consumed by children in care; and (2) Assess the association between centre healthy eating practices and child intake of fruit and vegetable servings, added sugar(grams), saturated fat(grams) and sodium(milligrams) in care. A cross-sectional study amongst 448 children attending 22 childcare centres in New South Wales, Australia, was conducted. Child dietary intake was measured via weighed lunchbox measurements, photographs and researcher observation, and centre healthy eating practices were assessed via researcher observation of centre nutrition environments. Children attending lunchbox centres consumed, on average 0.80 servings (standard deviation 0.69) of fruit and 0.27 servings (standard deviation 0.51) of vegetables in care. The availability of foods within children’s lunchboxes was associated with intake of such foods (*p* < 0.01). Centre provision of intentional healthy eating learning experiences (estimate −0.56; *p* = 0.01) and the use of feeding practices that support children’s healthy eating (estimate −2.02; *p* = 0.04) were significantly associated with reduced child intake of saturated fat. Interventions to improve child nutrition in centres should focus on a range of healthy eating practices, including the availability of foods packed within lunchboxes.

## 1. Introduction

Poor dietary behaviours, including low intake of fruit and vegetables, and a high intake of energy-dense discretionary foods (i.e., foods high in added sugar, sodium and saturated fat), are the leading modifiable risk factors for the development of obesity and non-communicable diseases [1]. As early childhood is a crucial period for the development of healthy eating behaviours, which are known to track into adulthood [2], improving the dietary behaviours of young children is recommended to reduce the burden of disease from obesity and non-communicable diseases [3]. Early childhood education and care (ECEC) is an attractive setting to deliver interventions to improve children’s dietary behaviours. At least 80% of children in Australia, United States (U.S.) and United Kingdom (U.K.) attend centre-based childcare, including long day care and preschools [4,5], providing multiple opportunities to influence and reinforce children’s healthy eating behaviours. Further, the fostering of children’s healthy eating behaviours aligns with the philosophy of the setting, as accreditation processes require childcare centres to create environments supportive of child health [6,7].

Within childcare centres that provide meals and snacks to children on site (i.e., menu-based centres), food availability and a number of additional characteristics of centre nutrition environments have been found to be associated with improved child dietary intake in care [8,9,10,11]. A recent review consolidating evidence from systematic reviews to examine the potential effectiveness of childcare centre practices concluded that healthy food availability was associated with improved child dietary intake in care [11]. A U.S. study conducted in 2000 with 28 children attending care concluded that positive educator role modelling of healthy eating increased child intake of fruit [8]. A 2015 study with 398 children from 24 Dutch preschools, found children ate more servings of fruit when they participated in meal preparation, and ate more vegetables when encouraged by educators to continue eating vegetables [9]. Further, a 2019 study with 58 preschool managers and 585 children in Finland found that having comprehensive written food policies which include educator feeding practices and the provision of food in care was associated with higher child intake of vegetables [10]. Given the association between centre nutrition environments and child dietary intake, best-practice guidelines recommend centres implement evidence-based healthy eating practices targeting the characteristics of centre nutrition environments known to be supportive of children’s healthy eating behaviours [12,13].

The majority of evidence to support the association between healthy eating practices, including food availability, and child dietary intake in care has, however, been conducted within menu-based centres [11]. At present, it is unknown whether such factors influence child dietary intake within centres where parents or guardians are required to pack food from home (i.e., within lunchboxes) for children to consume in care. Differing operational characteristics among menu-based and lunchbox centres may mean the associations between healthy eating practices and child dietary intake established in menu-based centres do not generalise [14,15]. For example, in Australia, lunchbox centres are more likely to cater only for children aged 3 to 5 years, and have shorter hours of operation compared with menu-based centres [14,15]. Furthermore, mealtimes within menu-based centres are becoming increasingly family style, with children encouraged to serve their own portions [13], compared to the pre-prepared portions available within children’s lunchboxes. Finally, sector-specific dietary guidelines exist for menu-based centres which specify the types and quantities of food to be provided to children in care [16,17], but do not provide such prescriptive guidance for the content of lunchboxes packed for parents. Despite lunchbox centres making up a significant proportion of Australian childcare centres [18], few studies have described the nutrition environment and healthy eating practices of lunchbox centres, nor examined the association between such factors and child dietary intake. Such evidence is required for the development of targeted interventions for this setting to improve public health nutrition.

In the context of the current evidence-base, the aims of this study were to: (1) Describe the nutritional content of foods and beverages consumed by children in care; and (2) Assess the association between centre healthy eating practices and child dietary intake of fruit and vegetable servings, added sugar (grams (g)), saturated fat (g) and sodium (milligrams (mg)) in care. It was hypothesised the implementation of centre healthy eating practices would be positively associated with child dietary intake of fruit and vegetable servings, and negatively associated with child dietary intake of added sugar, saturated fat and sodium in care.

## 2. Materials and Methods

### 2.1. Study Design and Setting

A cross-sectional study was undertaken in 22 Early Childhood Education and Care (ECEC) centres in the Hunter New England (HNE) region of New South Wales (NSW), Australia. ECEC centres included long day care and preschools, typically enrolling children between 0 and 6 years old, prior to compulsory schooling [19]. ECEC centres within the study region currently participating in a cluster randomised controlled trial (RCT) to improve child dietary intake in care served as the sampling frame for the study, with baseline data from the trial presented within this paper [20]. The sample size of 440 children and 22 ECEC centres (estimating 20 children per centre) was calculated to enable the detection of clinically meaningful differences in the trial outcomes [20].

Ethical approval to conduct the study was obtained from the HNE (approval no: 06/07/26/4.04) and the University of Newcastle (reference number: H−2008-0343) Human Research Ethics Committees.

### 2.2. Study Recruitment and Procedures

#### 2.2.1. Childcare Centres

Centres were eligible to participate in the study if they met the following criteria: (1) enrol more than 20 children per day; (2) have internet access at the centre; (3) do not provide meals or snacks to children (i.e., parents or caregivers provide food packed in lunchboxes); (4) not be currently participating in any other intervention to improve child healthy eating and/or physical activity; and (5) not be fully compliant with healthy eating practices targeted by the intervention and specified in the NSW state obesity prevention program (i.e., Munch & Move) according to NSW Ministry of Health data monitoring [21]. Centres were excluded from the study if they: (1) were a mobile preschool, family day care or a centre that did not provide care to children aged 2–5 years; (2) catered exclusively for children requiring specialist care; or (3) were run by the Department of Education due to differing operational characteristics.

Potentially eligible centres located within the HNE region were identified through data provided by the NSW Ministry of Health [21]. A recruitment package consisting of an information statement and consent form was progressively distributed to potentially eligible centres via mail and email in random order. Approximately two weeks later, a research assistant telephoned potentially eligible centres to review study details, assess centre eligibility, request consent for study participation and schedule a two-day data collection site visit with consenting centres. Centres were contacted in this random order until the required number of centres (*n* = 22) consented.

#### 2.2.2. Children

For children to be eligible, they were required to: (1) have prior written consent from a parent or guardian; (2) be aged between 2 and 5 years; and (3) not have a dietary restriction that requires specialised tailoring of their diet (e.g., allergies).

Following written consent from centres to participate in the study, centre staff were asked to distribute information statements and consent forms to parents with children aged between 2 and 5 years via usual communication methods with parents, including email, parent communication apps, and child pigeonholes. For those centres that consisted of more than one classroom, staff were asked to only distribute information statements and consent forms to the classroom with the highest number of children enrolled aged between 2 and 5 years. Prior to data collection and on the days of the site visit, trained research assistants (RAs) attended the centre during drop off time to discuss the purpose of the study with parents, assess child eligibility and obtain parent consent for child/ren to participate.

### 2.3. Data Collection Procedures and Measures

Data collection occurred between September 2019 and December 2019.

#### 2.3.1. Child Dietary Intake of Fruit and Vegetables Servings in Care

Measurements of lunchbox foods and beverages were conducted to assess servings of fruit and vegetables consumed whilst in care. Similar to lunchbox assessments previously conducted in childcare centres [22], two trained RAs attended each centre on two consecutive days to assess the lunchboxes of participating children. Lunchbox assessments were conducted over two days for a centre, but only on one day for each child. Specifically, lunchboxes were measured on two occasions across the day: before the first meal and after the last meal. During this process, RAs took a photo of the lunchbox contents and weighed each food item packed within the lunchbox, whilst adhering to strict food handling protocols. RAs then repeated this process after the last meal, with intake calculated based on foods and beverages present at the first measurement minus foods remaining at the second measurement. For food or beverage items that contained mixed ingredients (e.g., sandwiches, homemade baked goods, casseroles, pasta dishes), the RAs observed each food item and recorded a detailed description of each item, including name of item, estimated quantity (e.g., number of bread slices, cups of rice, tablespoons of sauce) and type (e.g., white, wholegrain). This approach has been taken for previous dietary assessments conducted within the ECEC setting [23]. All food wastage, including packaging, partially consumed items and food items dropped on the floor, were collected by the RAs throughout the day and were factored into child intake measurements. Weighed food record data is considered gold standard for measuring child dietary intake in the setting [20,22,24].

Prior to data collection, RAs completed a one-day training conducted by trained dietitians with experience in the data collection methods, in which RAs practiced weighing, observing and recording food and beverage items packed within children’s lunchboxes [22]. Prior to data collection, all RAs completed a practical assessment in which they were required to score above 80% on a test assessing their accuracy of weighing, observing and recording foods and beverages [22]. For quality assurance purposes RAs were accompanied by a trained dietitian with data collection experience for their first day of data collection.

Following data collection, a trained dietitian entered the weighed and observed food data into FoodWorks v10, a nutrient analysis database [25]. When the type or quantity of a food item was unclear, trained dietitians developed a list of standard assumptions to be applied across all lunchbox measurements. For example, a thin spread of margarine on a sandwich was assumed to be 0.5 teaspoons (2 g) of monounsaturated margarine, whilst one regular slice of cheese was assumed to be 21 g of reduced fat cheddar cheese. For quality assurance purposes, each assumption was checked by a minimum of two dietitians with experience evaluating child dietary intake within the childcare setting, with disagreements solved via consensus when required. All food and beverage items were categorised into food groups and serving sizes consumed in accordance with the Australian Guide to Healthy Eating (AGHE) to calculate the servings of fruit and vegetables [26]. When required, lunchbox photographs were used to validate food and beverage descriptions and weights recorded during data collection [22].

#### 2.3.2. Child Dietary Intake of Added Sugar, Saturated Fat and Sodium in Care

Child dietary intake of added sugar (g), saturated fat (g) and sodium (mg) from all food and beverage items consumed whilst in care was calculated using the nutrient output provided by the weighed food record data entered into FoodWorks v10 [25] following the process described above. Added sugar was defined as per the Australian Dietary Guidelines and Food Standards Australia and New Zealand (FSANZ), and includes sugars refined from plants [22,27]. Sodium was defined according to FSANZ, and includes the sodium content from all sources [27].

#### 2.3.3. Centre Healthy Eating Practices

A modified version of the Environmental and Policy Assessment and Observation (EPAO) tool [28] was used to assess the healthy eating practices potentially influencing child diet. The EPAO tool has been previously validated and is considered the gold standard in assessing childcare centre nutrition environments [28]. Selected items within the EPAO considered appropriate to the Australian lunchbox centre context were used to assess the following five practices: (1) supporting families to provide healthier foods consistent with dietary guidelines. For example, monitoring children’s lunchboxes and communicating with parents regarding lunchbox contents (two items); (2) provision of intentional healthy eating learning experiences, such as formal nutrition lessons and informal conversations (two items); (3) use of educator feeding practices that support children’s healthy eating. For example, educator role modelling healthy foods, avoiding the use of food as bribes and encouraging children to try new foods (21 items); (4) staff participation in professional development targeting healthy eating (one item); and (5) having a comprehensive written nutrition policy that outlines key healthy eating practices (13 items). Additionally, a sixth practice, the availability of food and beverages from foods packed within children’s lunchboxes, including fruit and vegetable servings, as well as added sugar (g), saturated fat (g) and sodium (mg), was calculated through the lunchbox measurement process described above.

As the EPAO was originally developed to assess menu-based childcare centres, items related to menu provision were replaced with items specific to lunchbox centres (e.g., educators monitoring children’s lunchboxes for compliance with dietary guidelines). Relevant EPAO items were identified and mapped to each of the healthy eating practices by health practitioners (i.e., a dietitian and public health nutritionist) with experience working with lunchbox childcare centres. The mapping of these items to each practice was then reviewed by two behavioural researchers (with experience in measurement development in the setting) to obtain consensus.

As per EPAO data collection training protocols [28], RAs completed a one-day training session conducted by a trained researcher with data collection experience prior to the conduct of data collection to familiarise themselves with the tool and data collection protocols. RAs also attended a childcare centre to complete a practice observation of a centre nutrition environment with an experienced member of the research team. During the practice observation, RAs and the research team member independently completed the EPAO and compared responses to ensure consistency in approaches. For quality assurance purposes, RAs were accompanied by a trained researcher with data collection experience for their first day of data collection.

In accordance with the EPAO protocol, on one of the two days of data collection, a trained RA completed a one-day observation (i.e., between 9am and 3pm) of the centre nutrition environment and reviewed childcare centre documentation [28]. The same room selected for the lunchbox measurements (i.e., the classroom with the highest number of children enrolled aged 2–5 years) also participated in the centre nutrition environment observation. The observation component of the EPAO assessed educator use of feeding practices supportive of children’s healthy eating, centres supporting families to provide healthier foods consistent with dietary guidelines and the provision of intentional healthy eating learning experiences. At each meal occasion (i.e., morning tea, lunch and afternoon tea), an RA observed the centre nutrition environment and recorded if a specific item was observed or not on the data collection form. The documentation component of the EPAO assessed centre nutrition policy contents, staff completion of professional development in nutrition, and evidence of supporting families to provide healthy foods consistent with dietary guidelines. An RA reviewed documentation and recorded the relevant contents of each document on the data collection form. Copies of relevant documentation, including centre nutrition policies and evidence of professional development, were collected to validate information recorded on the EPAO data collection form. Following the site visit, a trained RA entered the EPAO data collected via multiple sources (i.e., the observation and documentation review) into Excel. Items within each of the healthy eating practice were given a score out of three, with the score for each of the practices calculated by summing the scores from the relevant items, then dividing by the number of items within the practice (scoring range of 0–3). A score of 0 indicates a healthy eating practice is not implemented during any meal occasion (i.e., no item was achieved), whilst a score of 3 indicates a healthy eating practice is fully implemented at every meal occasion (i.e., all items were achieved).

#### 2.3.4. Centre and Child Demographics

Centre demographic information was collected during a telephone interview with centre nominated supervisors and included type of centre (i.e., long day care or preschool), days of operation, centre opening and closing hours, and number of children enrolled aged between 2 and 5 years [29,30]. Centre geographical information (i.e., postcode) was used to classify centre locality (i.e., either urban or regional/remote) and socio-economic status (i.e., either low or high social disadvantage).

Child demographics were captured through information recorded on parent consent forms. Parents reported their child’s age, sex (as recorded on the child’s birth certificate), usual number of days attending care, and child Aboriginal and/or Torres Strait Islander background.

### 2.4. Statistical Analysis

Statistical analyses were performed using SAS 9.3 software. Descriptive statistics, including means and standard deviations, were used to describe centre and child demographics, and the fruit and vegetable servings, sugar (g), saturated fat (g) and sodium (mg), consumed by children in care and packed in children’s lunchboxes.

Centre postcodes ranked in the top 50% of NSW according to the 2016 Socio-Economic Indexes for Areas (SEIFA) were classified as least disadvantaged (i.e., high SES), whilst the lower 50% of postcodes were classified as most disadvantaged (i.e., low SES) [31]. The Australian Statistical Geography Standard were used to classify centre locality as either urban or regional/remote [32]. Differences in centre SES and geographic location between consenting and non-consenting centres were examined via chi-square analyses to identify potential participation bias. Standardised scores were calculated for each EPAO item to account for the variation in the number of mealtimes within participating centres (i.e., preschools predominately had two mealtimes, long day care centres had three), allowing for direct comparison of healthy eating practices between centres with a different number of mealtimes. In the rare instances (1% of EPAO items) where data was missing for an EPAO item within a mealtime, we assumed the missing item had the same value as the recorded value for the other mealtimes for that centre. Multilevel mixed-effects linear regressions were performed to determine the association between overall and individual item healthy eating practices (independent variable) and measures of child dietary intake (dependent variable). These included a random intercept effect for the centre to account for potential clustering, as well as fixed effects for SES and centre locality to account for centre characteristics associated with child dietary intake. Statistical significance was defined as *p* < 0.05.

## 3. Results

Of the potentially eligible centres within the sampling frame (*n* = 85), 57 centres were sent an information statement and consent form. Of these, 25 (53%) centres declined to participate (lack of time, *n* = 21; study of lessor importance, *n* = 2; lack of staff capacity, *n* = 2) and 10 (21%) centres were ineligible (provided food to children, *n* = 3; NSW Department of Education centre, *n* = 6; involved in another healthy eating or physical activity study, *n* = 1), resulting in a study consent rate of 47%. There were no significant differences in centre area SES or centre geographic location between consenting and non-consenting centres. Within participating childcare centres, the average child consent rate to participate in lunchbox measurements was 75%, with lunchbox measurement data collected for 448 children (89.2% of consenting children due to absenteeism on data collection days).

The majority of participating childcare centres were preschools (90.1%) (Table 1) and enrolled an average of 29.9 (SD 9.8) children aged 2–5 years. Fourteen (63.8%) centres were located in high SES areas, with sixteen (72.7%) located in urban (major cities). On average, participating children were aged 4.7 years (standard deviation (SD) 0.7) and attended care for 2.6 days per week (SD 0.8).

### 3.1. Child Dietary Intake

Results of the lunchbox measurements indicate that children consumed a mean of 0.80 (SD 0.69) servings of fruit and 0.27 (SD 0.51) servings of vegetables (Table 2). Children consumed a mean of 8.06 g (SD 8.44) of added sugar, 5.57 g (SD 3.96) of saturated fat and 668.60 mg (SD 328.57) of sodium.

When examining the association between the availability of foods and beverages packed within children’s lunchboxes and child dietary intake, results of the multilevel mixed-effects linear regression indicate that there was a statistically significant association between fruit servings packed within lunchboxes and those consumed (estimate 0.51; standard error (SE) 0.02; *p* < 0.01) and vegetable servings packed and consumed (estimate 0.72; SE 0.02; *p* < 0.01) (Table 2). Results also indicated that there were statistically significant associations between the amount of added sugar, saturated fat and sodium available from foods and beverages packed within children’s lunchboxes, and child dietary intake of those nutrients (Table 2).

### 3.2. Association between Healthy Eating Practices and Child Dietary Intake of Fruit and Vegetables

The mean scores for centre healthy eating practices have been provided in Table 3. The highest mean score evident was the use of feeding practices that support children’s healthy eating with 1.86 (SD 0.22; range 1.57–2.36) out of 3. Results of the multilevel mixed-effects linear regression indicate there were no statistically significant associations identified between healthy eating practices and child intake of fruit and vegetable servings (Table 4).

Several statistically significant positive associations between individual items within the healthy eating practices and child intake of fruit and vegetable servings were identified (Table 4). Educators observing children’s lunchboxes for consistency with dietary guidelines was significantly associated with increased child intake of fruit servings (estimate 0.07; SE 0.03; *p* = 0.01), as well as educators using an authoritative feeding style (e.g., educators used supportive strategies such as reason and education, rather than bribes or threats) (estimate 0.09; SE 0.04; *p* = 0.04). Educators allowing children to choose between two healthy food options was significantly associated with increased child intake of vegetable servings (estimate 0.07; SE 0.04; *p* = 0.05).

The inclusion of several items within centre nutrition policies was significantly associated with child dietary intake. For example, the inclusion of educator participation in professional development in healthy eating was negatively associated with child intake of vegetable servings (i.e., intake decreased) (estimate −0.09; SE 0.04; *p* = 0.04), whilst the inclusion of staff avoiding the use of food to calm a child or as a bribe to get a child to behave was positively associated with increased child intake of fruit servings (estimate 0.11; SE 0.05; *p* = 0.02).

### 3.3. Association between Healthy Eating Practices and Child Dietary Intake of Added Sugar, Saturated Fat and Sodium

Results of the multilevel mixed-effects linear regression indicate that there were several statistically significant negative associations between the childcare centres healthy eating practices and child intake of saturated fat (i.e., it reduced the fat intake) (Table 4). This included centre provision of intentional learning experiences about healthy eating (estimate −0.56; SE 0.19; *p* = 0.01) and the use of feeding practices that support children’s healthy eating (estimate −2.02; SE 0.92; *p* = 0.04).

Multiple statistically significant associations between individual items within the healthy eating practices and child dietary intake of added sugar, saturated fat and sodium intake were also identified (Table 4). The provision of both formal (estimate −0.44; SE 0.16; *p* = 0.01) and informal (estimate −0.61; SE 0.22; *p* = 0.01) nutrition education to children and educators using an authoritative feeding style (estimate −0.50; SE 0.23; *p* = 0.04) were negatively associated with child intake of saturated fat (i.e., reduced intake). Having a variety of healthy foods visible to children during mealtimes was positively associated with child intake of added sugar (i.e., increased intake) (estimate 1.10; SE 0.43; *p* = 0.02), whilst educators requiring children to sit at the table until they finished food was negatively associated with child dietary intake of sodium (i.e., reduced intake) (estimate 127.55; SE 51.79; *p* = 0.02).

## 4. Discussion

To our knowledge, this is the first study to assess the association between healthy eating practices and child dietary intake within Australian lunchbox childcare centres. The study found low servings of fruit and vegetables consumed from foods packed within children’s lunchboxes, in addition to high intake of added sugar, saturated fat and sodium. The hypothesis that the implementation of centre healthy eating practices would be positively associated with child dietary intake of fruit and vegetable servings, and negatively associated with child dietary intake of added sugar, saturated fat and sodium was partially supported. The study found consistent associations between the availability of fruit, vegetables, added sugar, saturated fat and sodium in lunchboxes and child dietary intake of these in care, but less consistent associations between these measures of child dietary intake and other healthy eating practices. The findings provide important information for policy makers and practitioners interested in improving child nutrition in this sector.

Reported intakes of fruit (0.80 servings) and vegetables (0.27 servings) in this study is consistent with previous Australian studies examining child dietary intake within lunchbox centres [33,34]. For example, previous cross-sectional studies conducted within NSW have reported intakes of 0.7 servings of fruit and 0.1–0.2 servings of vegetables [33,34]. More broadly, cross-sectional studies conducted within U.S. childcare centres have also found low intakes of fruit and vegetables from foods packed within children’s lunchboxes to consume in care [35,36]. The limited intake of vegetable servings in care is of particular concern, with sector-specific guidelines recommending for children to consume one to two servings of vegetables whilst in care per day [12]. Such findings indicate that there is extensive scope to improve child intake of vegetables in care.

The availability of foods within lunchboxes were significantly associated with child dietary intake of these items (*p* < 0.01). This finding is consistent with previous research conducted within other education-based settings such as schools, where food availability was found to be closely associated with child dietary intake [37,38]. As the effectiveness of other supportive educator feeding practices are, to a large extent, reliant on healthy foods being available within lunchboxes, strategies to support parents lunchbox packing behaviours should be a priority. This is supported by a 2019 systematic review by Nathan et al. examining lunchbox interventions within schools and childcare centres, which highlighted that the inclusion of strategies that actively target parents was particularly important to improve child dietary intake in care [39]. Other Australian initiatives, such as the Healthy Lunch Box developed by Cancer Council NSW, have been established to provide specific evidence-based recommendations to support the packing of healthy lunchboxes [40]. However, the impact of such an initiative on parent packing and child dietary intake has not been evaluated to our knowledge.

Findings from the multilevel linear regressions suggest that in addition to improving availability, considerable opportunities exist to improve child dietary intake via other healthy eating practices. Despite the absence of a significant association between supporting families to provide healthier foods and child dietary intake of fruit and vegetable servings at an overall practice level, findings at the individual item level show promise. Educators observing children’s lunchboxes for consistency with dietary guidelines was significantly associated with increased child intake of fruit (estimate 0.07; SE 0.03; *p* = 0.01). As this is the first study to examine the association between healthy eating practices and child dietary intake within Australian lunchbox centres, this finding is particularly noteworthy. Future interventions aiming to improve child dietary intake within lunchbox centres should consider targeting this item in order to maximise the impact of the intervention.

The study found the provision of intentional healthy eating learning experiences was significantly associated reduced child intake of saturated fat (estimate −0.56; SE 0.19; *p* = 0.01). This is broadly in contrast to a cross-sectional study by Ward et al. examining the nutrition environments of 50 Canadian preschools, which found no association between the provision of intentional healthy eating learning experiences and total fat intake [41]. Our study findings of a statistically significant association between healthy eating learning experiences and reduced intake of saturated fat, but not for other nutrients considered as markers for energy-dense discretionary foods (i.e., added sugar and sodium), is noteworthy. The potential mechanism for this differential association should be explored further. Interestingly, our study did not find a significant association between centre nutrition policies overall and child dietary intake. This is in contrast to a study of menu-based centres by Lehto et al., which found that centres having a comprehensive written food policy resulted in higher child intake of vegetables in care [10]. The differences in reported associations between our study and those undertaken in menu-based centres may be due to a range of methodological differences between the studies. Alternatively, they may suggest the contextual differences between lunchbox and menu-based centres may alter the strength of association between these centre types. Further research is warranted to investigate such hypotheses.

Other than strategies targeting the availability of foods within children’s lunchboxes, a number educator-related healthy eating practices such as monitoring children’s lunchboxes, using an authoritative feeding style and the provision of intentional healthy eating learning experiences should be prioritised within future ECEC-based interventions. Given the potential impact on child dietary intake, strategies to support centres to implement these healthy eating practices are required. Employing theoretical frameworks, such as the Consolidated Framework for Implementation Research (CFIR) [42], can provide a systematic approach to identifying barriers to centres implementing the recommended healthy eating practices identified in this study, and develop support strategies accordingly. As this is the first study to examine the association between centre healthy eating practices and child dietary intake within lunchbox centres, additional studies with well-defined practices and validated items are required to confirm the healthy eating practices most influential on child dietary intake. Such evidence can provide guidance to practitioners to support centres in implementing healthy eating practices to improve child dietary intake in care.

### Strengths and Limitations

This study had several strengths, including the use of gold standard objective measures to assess both child dietary intake and centre healthy eating practices [24,28]. Additionally, the study obtained a high consent rate of 75% from parents for their child(ren) to participate in lunchbox observation and measurements. However, the study is not without its limitations. Whilst similar to other studies conducted in the ECEC setting [29,30], a consent rate of 47% was obtained from childcare centres to participate in the study, limiting the potential generalisability of study findings. The study eligibility criteria, including the exclusion of centres currently compliant with healthy eating practices specified within the NSW state obesity prevention program and centres run by the Department of Education, may further limit the potential generalisability of the findings. Dietary intake was measured across one day for each child, and therefore, does not take into account potential daily fluctuations in intake. Additionally, the inability of RAs to confirm the nutrient content of mixed ingredient food items packed within children’s lunchboxes (e.g., homemade baked goods) may have resulted in the under or over estimation of some nutrient values. As such, the results may not be a true indication of child dietary intake in care. Given the importance of adequate fruit and vegetable consumption and limited consumption of added sugar, saturated fat and sodium to achieve and maintain good health [43], future studies should explore child dietary intake of such food groups and nutrients and provide a comparison to relevant dietary recommendations [34]. Despite the use of gold standard methodology, the presence of an RA whilst observing centre nutrition environments may have unintentionally influenced centre staff behaviour that may have not otherwise occurred [41], such as role modelling healthy food choices or providing nutrition education to children. Additionally, centre nutrition environment observations were conducted by a single RA at each centre, with inter-rater reliability not formally assessed. Given a small sample size (*n* = 22 centres) and the large number of multilevel linear regressions performed, the results of the regressions should be interpreted with caution [44]. Finally, the cross-sectional study design precludes the assessment of casual relationships occurring.

## 5. Conclusions

Given this was the first study to examine the association between centre healthy eating practices and child dietary intake within Australian lunchbox centres, it contributes substantially to a previously limited evidence base. Findings of the study suggest that future interventions should focus on improving the availability of foods packed within children’s lunchboxes, in combination with targeting educator-related healthy eating practices to improve child dietary intake within lunchbox centres. Future research assessing child dietary intake and centre nutrition environments over multiple days within a broader range of childcare centres may be warranted to provide a better understanding of the association between centre healthy eating practices and child dietary intake.

## Figures and Tables

**Table 1 nutrients-13-01139-t001:** Demographic characteristics of participating centres and children.

Centre (*n* = 22)	*n*	%
Type of centre:		
Preschool	20	90.1%
Long Day Care	2	9.9%
Number of child enrolments aged 2–5 years (mean, SD)	29.9 (9.8)	-
Centre opening hours (mean, SD)	8 (0.9)	-
Number of days open per week (mean, SD)	4.9 (0.4)	-
Socio-Economic Indexes for Areas (SEIFA):		
Most disadvantaged (low socioeconomic status (SES))	8	36.4%
Least disadvantaged (high SES)	14	63.8%
Geographic location:		
Urban (major cities)	16	72.7%
Regional/remote (inner regional, outer regional, remote)	6	27.3%
Child (*n* = 448)		
Age (mean, SD):	4.7 (0.7)	-
Sex: Female Male	210 238	46.9% 53.1%
Aboriginal and/or Torres Strait Islander background	44	9.8%
Number of days attending care (mean, SD)	2.6 (0.8)	-

SD: standard deviation.

**Table 2 nutrients-13-01139-t002:** Servings and nutritional content for foods and beverages packed (available to children) within lunchboxes and consumed from children’s lunchboxes (*n* = 448), and the association between availability and child dietary intake.

Food Group or Nutrient	Packed within Lunchboxes	Child Dietary Intake	Percentage of Packed Consumed	Association between Availability and Child Dietary Intake
	mean (SD)	mean (SD)	%	Estimate (SE); *p* value
Fruit (serving)	1.33 (0.94)	0.80 (0.69)	60.15	0.51 (0.02); *p* < 0.01 *
Vegetable (serving)	0.40 (0.63)	0.27 (0.51)	67.50	0.72 (0.02); *p* < 0.01 *
Added sugar (g)	10.17 (10.37)	8.06 (8.44)	79.25	0.65 (0.02); *p* < 0.01 *
Saturated fat (g)	7.80 (5.12)	5.57 (3.96)	71.41	0.61 (0.02); *p* < 0.01 *
Sodium (mg)	917.42 (413.91)	668.60 (328.57)	72.88	0.00 (0.00); *p* < 0.01 *

SE: standard error; * Denotes a statistically significant association (*p* < 0.05).

**Table 3 nutrients-13-01139-t003:** Centre healthy eating practices (*n* = 22).

Healthy Eating Practice *	Mean Score (SD)	Range **
Supporting families to provide healthier foods consistent with dietary guidelines	0.62 (0.98)	0.00–3.00
Provision of intentional learning experiences about healthy eating	0.52 (0.99)	0.00–3.00
Use of feeding practices that support children’s healthy eating	1.86 (0.22)	1.57–2.36
Educator participation in professional development in healthy eating	0.32 (0.89)	0.00–3.00
Comprehensive written nutrition policy	1.02 (0.35)	0.35–1.96

* Each healthy eating practice was scored out of three. ** Represents the distribution of scores calculated across participating centres (i.e., lowest–highest score calculated within each practice). See Table 4 for availability of foods and beverage packed in lunchboxes.

**Table 4 nutrients-13-01139-t004:** Multilevel linear regression estimates of the association between centre healthy eating practices and child dietary intake.

Items	Fruit Intake (Serving)		Vegetable Intake (Serving)		Added Sugar Intake (g)		Saturated Fat Intake (g)		Sodium Intake (mg)	
	Estimate(SE)	*p* Value	Estimate(SE)	*p* Value	Estimate(SE)	*p* Value	Estimate(SE)	*p* Value	Estimate(SE)	*p* Value
Supporting families to provide healthier foods consistent with dietary guidelines										
Educator observed children’s lunchboxes	0.07 (0.03)	0.01 *	−0.00 (0.03)	0.91	0.00 (0.42)	0.99	−0.03 (0.16)	0.86	23.28 (17.94)	0.21
Centres provide feedback to families regarding lunchbox contents	0.00 (0.03)	0.94	−0.00 (0.04)	0.98	0.43 (0.46)	0.37	−0.32 (0.18)	0.09	−24.31 (20.03)	0.24
Overall practice	0.07 (0.04)	0.08	−0.00 (0.04)	0.93	0.32 (0.56)	0.58	−0.25 (0.22)	0.26	3.05 (25.25)	0.91
Provision of intentional learning experiences about healthy eating										
Formal nutrition education to children	0.03 (0.03)	0.28	0.03 (0.03)	0.43	−0.55 (0.41)	0.19	−0.44 (0.16)	0.01 *	1.70 (19.32)	0.93
Informal nutrition education to children	−0.00 (0.05)	0.97	0.07 (0.04)	0.10	−0.94 0.58)	0.12	−0.61 (0.22)	0.01 *	0.50 (27.87)	0.99
Overall practice	0.03 (0.04)	0.52	0.05 (0.04)	0.23	−0.78 (0.50)	0.14	−0.56 (0.19)	0.01 *	1.50 (24.08)	0.95
Use of feeding practices that support children’s healthy eating										
Educator used an authoritative feeding style	0.09 (0.04)	0.04 *	−0.05 (0.05)	0.28	−0.54 (0.63)	0.40	−0.50 (0.23)	0.04	21.86 (28.01)	0.44
Educator used food to calm an upset child.	0.28 (0.22)	0.22	−0.05 (0.24)	0.84	−3.99 (2.98)	0.20	−0.06 (1.21)	0.96	51.38 (137.63)	0.71
Educator encouraged children to sit	0.08 (0.04)	0.06	0.01 (0.04)	0.78	0.58 (0.55)	0.31	0.12 (0.22)	0.58	34.80 (24.03)	0.16
Educator let the children choose between two healthy food options	0.03 (0.04)	0.44	0.07 (0.04)	0.05 *	−0.13 (0.51)	0.80	−0.51 (0.19)	0.02 *	−11.50 (22.80)	0.62
Educator ate with the children during meal times	−0.01 (0.03)	0.79	−0.04 (0.03)	0.31	0.02 (0.46)	0.96	0.00 (0.17)	0.99	−22.97 (19.71)	0.26
Educator enthusiastically role modelling eating healthy foods	0.02 (0.04)	0.59	0.00 (0.04)	0.91	−0.27 (0.50)	0.59	−0.37 (0.18)	0.06	−11.58 (22.35)	0.61
Educator made fruit and veg easier to eat	0.00 (0.04)	0.93	−0.01 (0.04)	0.70	0.45 (0.47)	0.36	0.02 (0.18)	0.90	31.86 (20.16)	0.13
A variety of healthy foods are visible to children	0.03 (0.04)	0.47	0.02 (0.04)	0.63	1.10 (0.43)	0.02*	0.19 (0.19)	0.34	−9.52 (22.59)	0.68
Unhealthy snack foods are visible to children	−0.01 (0.04)	0.74	−0.06 (0.04)	0.12	0.61 (0.48)	0.22	0.18 (0.19)	0.35	30.18 (21.50)	0.18
Educator ate unhealthy foods during the meal time	−0.07 (0.13)	0.56	−0.16 (0.12)	0.22	1.85 (1.66)	0.28	−0.75 (0.65)	0.26	81.13 (73.52)	0.28
Educator shows indifference to children	0.08 (0.21)	0.71	0.19 (0.20)	0.35	−1.15 (2.70)	0.68	−1.47 (1.09)	0.20	−56.01 (119.31)	0.64
Educator insisted that a child eat a food	0.00 (0.05)	0.94	−0.02 (0.05)	0.77	−0.87 (0.67)	0.21	−0.55 (0.26)	0.05	−29.82 (30.28)	0.34
Educator negotiated with children to eat healthy foods	0.04 (0.05)	0.48	0.02 (0.06)	0.71	0.46 (0.75)	0.55	0.29 (0.28)	0.31	−40.42 (32.70)	0.23
Educator new or less preferred foods	−0.06 (0.06)	0.29	0.04 (0.06)	0.54	0.36 (0.76)	0.64	0.13 (0.30)	0.68	−49.07 (32.11)	0.14
Educator led pleasant conversations during meals	0.00 (0.04)	0.98	0.01 (0.05)	0.77	0.54 (0.59)	0.37	−0.22 (0.22)	0.34	−38.83 (25.85)	0.15
Educator praised children for finishing food	−0.05 (0.08)	0.48	−0.03 (0.08)	0.72	−2.07 (0.88)	0.03	−0.67 (0.37)	0.09	3.45 (46.70)	0.94
Educator reasoned with the children to eat healthy foods	0.00 (0.06)	0.98	−0.02 (0.07)	0.78	−0.39 (0.87)	0.66	−0.09 (0.34)	0.80	−61.61 (35.82)	0.10
Educator used food as a reward/ withheld food as a punishment	0.00 (0.06)	0.95	−0.08 (0.06)	0.20	−0.47 (0.79)	0.56	−0.39 (0.30)	0.22	33.45 (34.81)	0.35
Educator rushed children to eat.	0.01 (0.06)	0.84	0.08 (0.06)	0.18	−0.59 (0.80)	0.47	−0.46 (0.29)	0.13	−24.74 (35.33)	0.49
Educator required children to sit at the table until they finished all food	0.00 (0.10)	0.96	0.06 (0.10)	0.55	−0.87 (1.34)	0.53	−0.69 (0.47)	0.16	−127.55 (51.79)	0.02 *
Educator spoon fed a child	0.07 (0.14)	0.60	0.12 (0.14)	0.42	−1.93 (1.85)	0.31	0.08 (0.74)	0.91	105.49 (79.78)	0.20
Overall practice	0.19 (0.18)	0.31	−0.04 (0.19)	0.84	1.08 (2.50)	0.67	−2.02 (0.92)	0.04	−64.11 (111.48)	0.57
Staff participation in professional development in healthy eating										
Overall practice	0.00 (0.04)	0.95	−0.04 (0.04)	0.43	0.21 (0.61)	0.74	−0.06 (0.23)	0.79	22.86 (26.73)	0.40
Comprehensive written nutrition policy										
Encouraging children to eat healthy foods without bribes or threats.	−0.05 (0.05)	0.36	−0.05 (0.05)	0.37	−0.66 (0.72)	0.37	0.22 (0.27)	0.41	12.61 (32.39)	0.70
Avoiding the use of food to calm a child or as a bribe	0.11 (0.05)	0.02 *	0.04 (0.05)	0.44	−0.46 (0.72)	0.53	0.04 (0.27)	0.87	−4.24 (32.08)	0.90
Staff participation in professional development in healthy eating	−0.00 (0.04)	0.97	−0.09 (0.04)	0.03 *	0.70 (0.52)	0.20	0.04 (0.21)	0.85	40.25 (22.51)	0.09
Educators enthusiastically role model	0.10 (0.07)	0.17	0.00 (0.07)	0.96	−0.73 (0.96)	0.46	−0.01 (0.37)	0.98	40.39 (42.16)	0.35
Providing a planned nutrition education activity for children	−0.03 (0.11)	0.80	−0.15 (0.11)	0.18	1.34 (1.50)	0.38	0.39 (0.55)	0.48	4.24 (69.14)	0.95
Checking with a child about their hunger/fullness before removing food	0.02 (0.05)	0.67	−0.04 (0.06)	0.48	0.64 (0.72)	0.38	0.30 (0.27)	0.28	5.39 (33.67)	0.87
Talk with children about food and provide informal nutrition education	0.05 (0.04)	0.30	0.03 (0.05)	0.54	−0.65 (0.59)	0.28	0.00 (0.23)	0.98	26.14 (26.32)	0.33
Strategies are in place to ensure that food brought from home is consistent with Australian Dietary Guidelines	0.01 (0.05)	0.90	0.02 (0.06)	0.69	−0.33 (0.73)	0.66	−0.21 (0.27)	0.45	−3.34 (32.90)	0.92
Offering families education on child nutrition once or more times per year	−0.02 (0.05)	0.74	0.03 (0.05)	0.57	−0.27 (0.64)	0.68	−0.13 (0.24)	0.60	31.43 (28.13)	0.28
Educators making positive comments about healthy foods eaten by children	0.05 (0.04)	0.30	0.03 (0.05)	0.54	−0.66 (0.59)	0.28	0.01 (0.23)	0.98	26.14 (26.32)	0.33
Praising children for trying new or less preferred foods	0.03 (0.05)	0.58	−0.02 (0.06)	0.70	−0.82 (0.71)	0.26	−0.20 (0.28)	0.47	37.94 (31.24)	0.24
Educators avoid using preferred foods to encourage children to eat new or less preferred foods	−0.08 (0.05)	0.13	0.00 (0.06)	0.95	−0.64 (0.75)	0.41	−0.18 (0.29)	0.55	−7.02 (34.03)	0.84
Educators not eating unhealthy foods or unhealthy beverages	−0.07 (0.06)	0.25	−0.03 (0.06)	0.64	0.27 (0.85)	0.75	−0.01 (0.32)	0.98	−8.59 (37.50)	0.82
Overall practice	0.05 (0.11)	0.63	−0.04 (0.11)	0.72	−1.04 (1.49)	0.50	−0.01 (0.57)	0.99	80.72 (64.32)	0.23

* Denotes a statistically significant association (*p* < 0.05).

## Data Availability

The data presented in this study are available on request from the corresponding author.

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
