# Peer review of "The Association between Australian Childcare Centre Healthy Eating Practices and Children’s Healthy Eating Behaviours: A Cross-Sectional Study within Lunchbox Centres"

_nutrients, 2021, doi:10.3390/nu13041139_

Round 1
Reviewer 1 Report
This paper presents the results of a cross-sectional study exploring the content of lunchboxes and the nutrition-related environment in ECECs. The paper is incredibly well written and presents a clear overview of the study and its findings. The introduction, in particular, is very well written and provides a strong and succinct rationale for the study objectives. Studies of food from home are especially needed today, as more programs shift away from food prep due to COVID-19 concerns. There are a number of smaller edits needed (see below for list). The most consequential questions I have concern how you translated pictures and observations into nutritional quality without knowledge of food preparation.
Abstract: within the word limits, it would be great to add a bit more detail about the methods
Methods:
- Provide a justification (power calculation? budget limitations?) for the n=22 sample size
- If possible, provide more information on how pictures/weights were translated into servings and contents, especially for "homemade" or "combination" items (e.g. casserole, leftovers). How did the RAs confirm the contents of items in the lunchbox? This is especially problematic for the estimation of sodium consumption, which is highly dependent on the method of preparation (e.g. added salt and/or fats during preparation).
- Some issues with parenthetical use in EPAO section (p4, lines 180-186).
- Provide a brief summary of the training provided to the RA prior to administering the EPAO (e.g. certification? observed/verified by second observer?). The accuracy of the single reporter is related to the validity of these data for your study.
- What about foods dropped on the floor or spilled? Could you confirm that these were also collected by RAs?
- Confirm how the two days of data collection were integrated into data analysis: were the 2 days of lunchbox contents averaged for each child? Were children with only one day of data included? if so, how?
Results:
- Table 1: I'm unsure what the mean value is for Aboriginal background - if this represents the mean number of children per center with this background, it belongs in the centre section and if this represents the number of children with this background, it should be listed as n, % consistent with the sex variable.
- Table 2: It would be helpful to also present the percentages consumed (e.g. 0.80 / 1.33 for fruit) to more clearly compare the food groups/nutrients
- Table 3: If the items were scored out of 3, how could the ranges be 1.57-2.36 or 0.35-1.96?
- Table 4: I suggest moving the last rows (relationship between packed and consumed) to Table 2 since it does not relate to the practices of the rest of the table. Then you could also move mention of this relationship to the dietary intake section - basically confirming that the more they are packed, the more they eat (as you say in Discussion).
Discussion:
- I suggest reordering this section to start with availability before intake - as you say in the text, they can only consume what is available.
- In lines 393-395 you say that "..few centres implement [the] healthy eating practices". This is slightly misleading as you specifically recruited programs that were not already following recommendations (p3, lines 105-106).
- 4.1 limitations: your methods says you observed 2 days (p4 line 139) and here you say one day. Please reconcile.
- Additional limitations are: 1) your stringent eligibility criteria limits generalisability: you excluded programs that were following guidelines, state affiliated, etc. This should be mentioned explicitly here. 2) you did not confirm the preparation of foods with parents so your estimates of fat and sodium may be inaccurate.
Reviewer 2 Report
An interesting study assessing the relationship between various practices for maintaining healthy nutrition in Australian lunchbox childcare centres and the consumption of selected products (beneficial for health) or ingredients (harmful to health in excessive amounts) by children. Unfortunately, neither the title nor the keywords contain the most important words "lunchbox centers" or lunchboxes prepared by parents.
Other comments to the article to consider:
- suggestion to write the title as an answer to the hypothesis,
- keywords need to be analyzed/changed,
- suggestion to provide research hypotheses at the end of the Introduction chapter,
- it should be clarified what does "added sugar" or "sodium" (NaCl?) mean? Was salting of products included in the research?
- in this type of research it is also very important how many children (what percentage) consumed too little portions of fruit and vegetables, or too much sugar or sodium, did the authors assess it in the project?; too much fruit consumption may also be disadvantageous due to the content of simple sugars,
- line 208-210 - "Items within each of the healthy eating practice were given a score out of three (...) (scoring range of 0-3) "- can you describe what the number 0 means (or 3),
- in the discussion, I suggest first of all to provide information about the confirmation of the hypothesis / hypotheses, and then divide the discussion into sections (adequately to the description of the results) - it will make the chapter more readable - the description of the results should not be repeated,
- the conclusions need to be improved.
Round 2
Reviewer 2 Report
Accept in present form.